# Comparative Toxicity of Interferon Beta-1a Impurities of Heavy Metal Ions

**DOI:** 10.3390/medicina58040463

**Published:** 2022-03-23

**Authors:** Dmitriy Berillo

**Affiliations:** 1Department of Pharmaceutical and Toxicological Chemistry, Pharmacognosy and Botany School of Pharmacy, Asfendiyarov Kazakh National Medical University, Almaty 050000, Kazakhstan; berillo.d@kaznmu.kz; 2Atchabarov Research Institute of Fundamental and Applied Medicine, Almaty 050000, Kazakhstan

**Keywords:** gel electrophoresis, interferon beta-1a, heavy metals, protein standardization, protein-metal complex

## Abstract

*Background and Objectives*: Providing a proper quality control of drugs is essential for efficient treatment of various diseases minimizing the possible side effects of pharmaceutical active substances and potential impurities. Recent in vitro and in vivo studies have shown that certain heavy metalloids and metals interfere with protein folding of nascent proteins in cells and their biological function can be altered. It is unknown whether the drug impurities including heavy metals may affect the tertiary protein structure. *Materials and Methods*: ReciGen and Rebif are pharmaceutical interferon beta-1a (IFNβ-1a) contained in preparations that are used for parenteral administration. Heavy metal impurities of these samples have been studied by gel electrophoresis, Fourier-transform infrared spectroscopy (FTIR) and inductively coupled plasma mass spectrometry analysis (ICP MS). The concentration of heavy metals including mercury, arsenic, nickel, chromium, iron, and aluminum did not exceed permitted levels established by International Council for Harmonisation guideline for elemental impurities. *Results:* The ICP MS analysis revealed the presence of heavy metals, moreover zeta potential was significantly different for IFNβ-1a, which can be an indirect indication of the difference in composition of ReciGen and Rebif samples, respectively. FTIR analysis revealed very similar amide I and II bonds at 1654 and 1560 cm^−1^ attributed to the peptide absorption peaks of IFNβ-1a in Rebif and ReciGen. *Conclusions*: It was hypothesized that the IFNβ-1a complex binds heavy metals affecting the tertiary protein structure and may lead to some side effects of drug administration. Further testing of IFNβ-1a bioequivalence for parenteral application is necessary.

## 1. Introduction

Interferons (IFNs) are a large family of related cytokines with the ability to confer resistance to viral infections and are used as immunostimulants in short-term treatments, while IFNβ is administered to treat multiple sclerosis requiring long-term prescription [1,2]. Some reports indicate that IFNβ-1b helped treat SARS-CoV-2 in the early stage of the disease development up to 7 days [3,4,5]. An immune system activation throughout the whole body is not desirable, therefore, local administration is more desirable. Thus, the nasal form is the most effective IFN application for COVID-19 treatment due to rapid pharmacokinetics for intravenous and parenteral administration despite numerous side effects [6,7]. IFNβ-1a is available only as an injectable form and can causes side effects such as skin reactions at the injection site including cutaneous necrosis, local destruction of fat tissue; symptoms of influenza infections such as fever, muscle aches, fatigue, and headaches lasting several hours after administration. Certain side effects such as fever and pain can be reduced with over-the-counter nonsteroidal anti-inflammatory drugs. The treatment of multiple sclerosis requires a break of approximately 48 h after administration that can prolong from a year to decades until recovery [8]. Drugs including IFN contain different heavy metals with their concentrations at permitted levels established by the International Council for Harmonisation (ICH) guideline for elemental impurities [9]. However, the cumulative effect of drug impurities should be accounted for. Different profiles of impurities can induce various impacts. For example, compounds of mercury can lead to neurological and behavioral disorders; chromium intoxication effect on increase of carcinogen and mutagen cases; arsenic is carcinogenic and may trigger cardiovascular disease [10,11]. It is known that IFNβ-1a is composed of several protein domains and two domains linked via coordination bond of zinc atom [12]. A comprehensive study on the influence of heavy metals to changes in the self-assembly of α-helical coiled coils in aqueous solution is discussed [13].

A great number of IFN-containing pharmaceutical forms are in clinical trials. It was estimated in January 2021 that 172 clinical trials involving IFN-based preparation usage were at diverse stages of accomplishment: two were in early phase I, 50 in phase I, 70 in phase II, 28 in phase III, and six in the final phase IV of clinical trials [14]. The certification and registration of the product proceeds by facilitated way, the comprehensive Food and Drug Administration (FDA) approval process is described elsewhere [15]. Two drugs are ‘bioequivalent’ as distinct by the World Health Organization (WHO), if it contains the same active pharmaceutical ingredient (API) in the same dosage form, equivalent strength and path of administration, and their bioavailabilities in term of pharmacokinetics, pharmacodynamics, in vitro, in vivo and clinical trials. For example, only 43% of the 2336 generic drugs containing an API were legally checked on bioequivalence due to a low number of certified laboratories in Chile. This issue with bioequivalence occurs very often in developing countries [16]. An additional supplementary substance can be added to preparations to produce a generic version with improved pharmacokinetics or longer storage time due to inhibition of oxidation, denaturation, etc. Efficient therapeutic delivery strategies of various proteins are based on smart polymers [17,18,19]. There are several polyethylene glycol (PEG)-conjugated forms of glycosylated recombinant IFNβ-1a such as Pegasys^®^, PegIntron^®^, ViraferonPeg^®^, and Plegridy Plegridy^®^ varying by the length of a single, linear molecule of 12–40 kDa. PEG-*O*-2-methylpropionaldehyde is known for patient treatment with relapsing forms of multiple sclerosis due to the decreased clearance [17]. Different brand supplies with chemical reagents may affect the drug properties. Even if a company produces a generic drug with the same reproduced technical quality, the product can alter from the original due to interactions with different levels of impurities. To be competitive with the preparation price, companies may change suppliers for drug production, yet not foresee that a slight change of precursor or solvent quality for purification may lead to a significant difference in purity of the final drug. In a recent review, different multistep biotechnological strategies of interferon production were discussed. Thus, for example, IFNβ can be produced with a yield ranging from 0.255 to 12 g/L [17]. Chromatographic techniques involving ion-exchange, size exclusion chromatography, or hydrophobic interaction chromatography are utilized to gradually decrease the concentration of denatured proteins. For example, PEGylated IFNβ-1a is administered subcutaneously [17], while injectable IFNβ-1a forms bear increased requirements for purity including sterility, restrictions on related impurities, the absence of progeny and bacterial endotoxins, heavy metals, etc. [20]. Protein complex formation with heavy metals affects the 3D protein structure [21] and leads to side effects of drug administration such as increased risk of developing neuronal damage in many neurodegenerative disorders, including Alzheimer’s disease and amyotrophic lateral sclerosis [11,22]. Moreover, heavy metal complexes are responsible for the generation of reactive oxygen species [11]. However, the effect of metal ions for IFNβ form stability have not been identified yet.

The aim of the current study is to compare two brands of human recombinant IFNβ for intervention administration on the content of metal ions. The quality and quantity of IFN was estimated using sodium dodecyl-sulphate (SDS)-gel electrophoresis and high-performance liquid chromatography (HPLC) analysis.

## 2. Results and Discussion

The mechanism of action of IFNβ appears to increase production of anti-inflammatory agents and to decrease the expression of pro-inflammatory cytokines in the brain diminishing the number of inflammatory cells across the blood–brain barrier [23]. Therefore, such a drug must be free of potentially toxic impurities. The IFNβ-1a protein weight of 2 of the Rebif and ReciGen samples were measured by gel electrophoresis (Figure 1). It is known that the molecular weight of IFNβ-1a is 22.5 kDa, [24] and these data well correlate with the studied Rebif and ReciGen samples. There were no additional stains on the acrylamide SDS gel which is an indication of monocomponent preparation.

As and Hg were detected in the ReciGen sample, but not in the Rebif sample (Table 1). According to the ICH Q3D chapter, impurities of elements, mercury and arsenic are attributed to the first class of toxicity [25]. As bears no known proper biological function in humans and has a 95% absorption rate by the body [9]. Genotoxicity of mercury and its derivatives demonstrated in vitro and in vivo in human populations studies were shown recently in a review by [26]. Toxicity of organic and inorganic mercury species was detected in differentiated human neurons and astrocytes as well as affecting liposomal integrity at concentrations ranging from 0.1–1.0 μmol [27]. Considering these results, additional control experiments on chronic toxicity must be performed to confirm the safety of application for patients with chronic disease treatment including sclerosis. Thus, the acceptable daily consumption exposure element impurity in drugs for arsenic is 15 μg/day, which in recalculation to one syringe of IFNβ-1a contains 0.0013 μg of arsenic and is therefore considered as acceptable. On the other hand, the permissible daily exposure to the impurity of element mercury in inorganic form for drugs with parenteral exposure equates to 3 μg/day [9]. Hence, the 0.5 mL of ReciGen syringe with 0.0005 μg/day of mercury is apparently a safe dose. However, the ICH guideline for elemental impurities Q3D (R1), safety assessment and derived permitted daily exposure does not apply to organic Hg [9]. More studies are necessary to confirm the safety of the IFNβ-1a in this case.

IFNβ-1a contains a complex structural formula C_908_H_14_O_8_N_246_O_252_S_7_ with a molecular weight of 22.5 kDa [24]. The macromolecule structure contains seven sulphur atoms, and many carboxyl and amino functionalities, therefore, bearing a high affinity for binding to heavy metals [28,29] including Hg [21]. The protein binding with heavy metals can trigger the change of the native confirmation according to numerous reports [21,28,29,30]. Ni is the element attributed to the class 2A found in amounts 2.7 times more in ReciGen vs. Rebif (Table 1). The pharmaceutical preparations of nickel contain concentrations of 13.8 and 4.63 μg/L, respectively. The daily exposure for parenteral exposure is 20 μg/day. Consequently, one syringe of the drug with a volume of 0.5 mL contains up to 0.002 μg of Cr. Cr belongs to the 3rd hazard class. The Rebif sample exhibited better purity and quality than the ReciGen sample. It is worth highlighting that there is an established interaction between INFα and 1 mM Zn concentration that forms. Ni and Zn bind to glutamine and histidine residues of proteins. The biological role of the INFα dimer is currently unknown [30]. It is not known whether IFNβ-1a form dimers through Zn. Any dimers were not observed for IFNβ-1a by gel electrophoresis as shown previously; however, it may be restricted by the sensitivity of the method (Figure 1).

In some cases, lower levels of established element impurities even below the toxicity thresholds may affect qualitative characteristics of the medicinal product (for example, if element accelerates the degradation of active substances, etc. [9].

According to the heavy metals analysis, the IFNβ-1a pharmaceutical preparation of Rebif revealed better purity results than the ReciGen sample. It is known that proteins have a high affinity to the most toxic d-elements and can form strong chelation complexes. Binding with heavy metal cations can lead to active center blocking or conformation changes [31]. That is why, IFNβ-1a may not be as pharmacologically active as formulations that are free of metals. Thus, according to calculations, the ReciGen drug contained 0.88 nmol of heavy metals, that is 45 molar % of IFNβ-1a can be chelated to the protein, therefore, it is probable that this drug does not have a studied and confirmed biological activity. Meanwhile, the Rebif sample had permitted concentrations of Ni and Cr. Nevertheless, accounting for the molar content of these elements relative total content of IFNβ-1a, there are about 4.4 molar % of metal ions relatively of IFNβ-1a.

Aluminum was found in the pharmaceutical preparation at a concentration of 7.40 ± 0.17 μg/L. Metal ions especially multivalent cations form complexes with various protein structures. It was shown that Al ion reaction with blood compounds led to the immediate iron transporter transferrin and human serum binding with Al [32]. It was suggested that metal-induced autoimmunity had occurred, i.e., the Al usage as vaccine adjuvant led to complement activation and could interact with antigen presenting cells. The subsequent immune response could contribute to neuronal damage and β-amyloid deposition [33].

To confirm the hypothesis of IFNβ-1a metal ion complex formation, the structure complexes have to be established using SAXS, microcalorimetry. It can be detected by an X-ray structural method and HPLC with a mass detector to determine the IFN location and composition with heavy metals via molecular ion decay and fragmentation. IFNβ-1a (Rebif) and IFNβ-1a (ReciGen) samples had a characteristic peptide-bond frequency peak at 1654 cm^−1^ (Figure 2). According to the research data, IFNβ-1a had a frequency of the amide I bond at 1650 cm ^−1^ confirming the alpha-helices predominance in the structure [34]. In general, both FTIR spectra are very similar, and the difference is only in intensity, the peak at 1562 cm^−1^ corresponds to amide II bond of peptide. Some FTIR spectrum differences with data from the literature may be associated with a different spectrum recording mode or different profile of impurities in the pharmaceutical preparations. Other differences in FTIR spectra may be attributed to some variations of supplementary substances that can be investigated in detail using HPLC analysis. FTIR analysis confirms the presence of peptide in native form in both samples. More detailed information cannot be extracted from these spectra due to presence of supplementary substances such as acetate, methionine, poloxamer, mannitol due to overlay of signals of functional groups. Circular dichroism method [34] can be used for the future studies and comparisons of the tertiary structure of IFNβ-1a and with standard IFNβ-1a sample ca.

Taking into account the specification of both formulations presented in Table 2, it has identical chemical composition and thus expected similar physicochemical characteristics. Analysis of zeta potential ReciGen and Rebif solutions of INF illustrated 36.2 ± 2.5 and 25 ± 6.6 mV, indicating that it is a positively charged protein at pH 4.2 (Figure 3). The difference in values of zeta potential may be related to the presence of impurities of metal ions. It was impossible to obtain a good quality stable signal for DLS analysis to evaluate the average size of the protein due to measurements performing lower than limit of detection (Data are not shown). In general, at sufficient protein concentrations using DLS one can expect detection of some dimers and protein aggregates.

To facilitate perception of complexity of the 3D structure of the IFNβ-1a the software Mol* Viewer was used for 3D visualization and highlighting of the zinc-binding site holding two protein domains (Figure 4). The IFNβ-1a has three hydrogen bonds between Tyr_125_ OH⋅⋅⋅Asn-1 Nδ2, Tyr_126_ OH⋅⋅⋅Asn_153_ Oδ1, and Tyr_125_ OH⋅⋅⋅Glu_149_ Oε2 facilitating interactions between helices D and E. In addition, a zinc ion is observed to exist at the interface between domain A and B, coordinating in a tetrahedral manner via His_121_ of molecule A and His_93_ and His_97_ of domain B, meanwhile water molecules associate to the 4th coordination site. A system of hydrogen bonds formed between His_121_ and Glu_43_ (domain A) and between His-_97_ and Gln_94_ (domain B) appears to assist in the stabilization of the zinc-binding site [12]. Presence of extra concentration of polyvalent ions may interfere with the formation of hydrogen bonds and stabilization of the complex [35]. A very recent study illustrated that copper binding near the N-terminus led to a net unfolding of 6–7 residues, that was proven by CD spectra and protein stability evaluation [36]. Further study will be devoted to estimation of half-life time of IFNβ-1a in the presence of heavy metals.

## 3. Materials and Methods

PageRuler™ plus prestained protein ladder (Thermo Fisher Scientific, Kingfisher Dr, Swindon, UK), Coomassie blue R250, Tris *HCl, TEMED 99% were purchased from Sigma Aldrich (Darmstadt, Germany); acrylamide 99.9%, methylenbisacrylamide 99.5%, glycine 99.5%, dodecylsulphate sodium 99.9% were obtained from LLP LaborPharma (Almaty, Kazakhstan).

PerkinElmer Pure Plus Atomic Spectroscopy standard N9300235 Multi-element Calibration Standard 5 Matrix H_2_O/0.2% HF/HNO_3_ LotN CL49-257CRY1, Multi-element Calibration Standard 1 Lot N 6-95MKBY2, Agilent technologies.

IFNβ-1a Rebif and ReciGen were received from Merck company (Tashkent, Uzbekistan) by chief freelance neurologist of the Republic of Kazakhstan prof., M.D. Turuspekova ST (Society of Disabled Patients with Multiple Sclerosis, Nur-Sultan, Kazakhstan), who kindly provided valid and properly stored samples for the current study. A limitation of this study is an absence of a certified IFNβ-1a reference standard (CRS(Y0001101) usually used for physicochemical characterization and confirmation of the origin of protein and Interferon beta (Human, rDNA, glycosylated), NIBSC, code:00/572) and used for biological tests. However, commercially available IFNβ-1a samples passed the standardization process and were utilized for the current study. It can be concluded that both proteins are of approximately the same molecular weight with the same values of hydrodynamic radius (Figure 1).

The United States Pharmacopoeia (USP) is an independent scientific organization that develops quality standards for medicines, dietary supplements and food ingredients. EurPh, European Pharmacopoeia; IFNβ-1a, interferon β-1a.

### 3.1. FTIR Spectroscopy

FTIR spectra of IFNβ-1a were recorded in the range of 4000–600 cm^−1^ with a resolution of 0.5 cm^−1^ and 1.0 cm^−1^ on a Cary 660 FTIR (Agilent technologies, USA), respectively. The IFNβ-1a solutions were frozen at −20 °C for 6h and then freeze-dried overnight. The received powder was stored in dry state under vacuum in dark conditions until the recording FTIR spectra.

### 3.2. Gel Electrophoresis

Electrophoresis was performed on a polyacrylamide gel cross-linked with 40: 1 methylenbisacrylamide with sodium dodecyl sulphate and measured by electrophoresis system Mini-PROTEAN^®^ Tetra cell (Bio-Rad, Gladesville, Australia) according to a previously described protocol [38]. The IFNβ-1a solution was heated at 100 °C for 5 min and electrophoresis under non-reducing conditions (20 ng IFNβ-1a protein-loaded or PageRuler™ plus prestained protein ladder (Thermo Fisher Scientific)) using 4% Tris hydrochloride [39]. Gradient polyacrylamide gels (Bio-Rad Ready Gels) for 1 h. Eighty μL of IFNβ-1 and 20 μL of 5× Sample Buffer solution, in each well contained 2.63 μg of IFNβ-1a Rebif or ReciGen that corresponded to 25 μL of solution. The electrophoresis was carried out applying 40 mA with voltage of 150 V. The staining of proteins was carried out using Coomassie blue R250. A more precise technique is described in the source [39].

### 3.3. Inductively Coupled Plasma Mass Spectrometry (ICP-MS)

IFNβ-1a samples were analyzed using ICP-MS in the Department of Chemistry at Al-Farabi Kazakh National University (Almaty, Kazakhstan). The used mass spectrometer with inductively coupled plasma Agilent 7500 (Agilent Technologies, USA) was calibrated with a standard solution (N9300235) with multi-element calibration standard 5 matrix H_2_O/0.2% HF/HNO_3_ (CL49-257CRY1), multi-element calibration standard 1 (6-95MKBY2, Agilent Technologies) containing Al, As, Ba, Be, Cd, Co, Cr, Cs, Cu, Fe, Ga, Li, Mn, Ni, Pb, Rb, Se, Sr, Tl, U, V, Zn, Hg according to the manufacturer’s protocol. Mass measurements ranged from 2 atomic units up to 260 atomic units. The permissible RMS value (repeatability) limit in a series of successive determinations of the peak areas did not exceed 2% and the measurement uncertainty was ±2.3%. The results of three experiments were averaged, and the results were obtained for each element, and the detailed method was described earlier [38,39].

### 3.4. Zeta Potential Analysis

The average zeta potential of the IFNβ-1a ReciGen and IFNβ-1a Rebif was evaluated without preliminary sample preparation (0.088 mg/mL of IFNβ-1a in acetate buffer pH 4.2) using a zeta potentiometer (Zetasizer 3000, Malvern Instruments, Warriewood, Australia). The following setting used 5 scans for each run of the sample and 3 runs were collected. The experiment was repeated at least 6 times. DLS analysis was performed using Zetasizer 3000 (Malvern Instruments, Warriewood, Australia). The size distribution was determined applying 10 scans for each run of the spectrogram and at least 10 runs were recorded.

### 3.5. Calculation of Molar Components in the Pharmaceutical Preparation of IFNβ-1a

Each injection dosage unit of IFNβ-1a contained 44 mg. ICP-MS data reveal that IFNβ-1a ReciGen sample showed an acceptable amount of Hg (0.0025 nmol), Zn (0.0225 nmol), Ni (0.04 nmol), Fe (0.655 nmol), Cr (0.132 nmol), Mn (0.0127 nmol), and Al (0.0137 nmol). Rebif sample showed an acceptable amount of Ni and Cr contained in IFNβ-1a Rebif, which were 0.0146 nmol and 0.0711 nmol, respectively.

A relative percent of IFNβ-1a with metal ions was calculated according to Equation (1):ω (%) = Σ mole (metal ion)/mole (of IFNβ-1a) × 100%(1)

The sum of heavy metal moles in the ReciGen and Rebif were 0.88 nmol with 45.05% of ω value and 0.0857 nmol with 4.4%, respectively.

Foreseen accumulative exposure dosage after IFNβ-1a ReciGen administration (44 μg in 0.5 mL) for one year was calculated according to Equation (2):γ (mole) = 365 days × 3/7 × mole (metal ion in the syringe)(2)

It is a general practice that patients receive IFNβ-1a every 48 h and in average 3 days a week.

## 4. Conclusions

According to the impurity parameters in drugs regulated by ICH Q3D guidelines, the guidelines for elemental impurities in both IFNβ-1a pharmaceutical preparations does not exceed the content of individual free metal ions. It is rather concerning that 1st class toxicity metals were detected in the pharmaceutical preparation for parenteral administration. Mercury and arsenic were identified at concentrations of 1.05 and 2.69 μg/L, respectively. Gel electrophoresis did not reveal a significant difference between the two brands, i.e., no protein aggregation. However, in this particular case, due to the high molecular weight of IFNβ-1a and relatively low molecular weight of metals, the molar ratio is quite equal. It is worth noting that at least one metal ion interacts with 2–3 carbonyl groups and several available amino components forming donor–acceptor bonds can result in a change of conformation of the native structure. According to the significant difference in zeta potential between IFNβ-1a Rebif and IFNβ-1a ReciGen, this may be attributed to the binding of metal ions and the native biological activity even though it does not exceed the established limits of heavy metals, i.e., their use will probably not produce significant side effects due to heavy metals. Nevertheless, R&D pharmaceutical companies must control the content of metals in the composition of injectable protein-based preparations due to their high ability for chelation, leading to the alteration of the native tertiary structure of many proteins, which in turn leads to aggregation of macromolecules or loss of therapeutic efficiency. The IFNβ-1a Rebif contains approximately 4.5 molar % of ions relative to the protein amount, but IFNβ-1a ReciGen contains approximately 10-fold more, which may be an acceptable level according to the ICH Q3D. In this regard, at the stage of the standardization process of protein-based drugs, it is necessary to carry out bioequivalence tests and to check for safety regarding chronic toxicity, since a chelated complex of interferons or another pharmacologically active protein with metals may bear different pharmacological activity. A comprehensive analysis of the identification and quantification of interferons in the two practices by brands ReciGen and Rebif can be performed using RP HPLC analysis with a mass detector supplied with an RP C4 column, circular dichroism and X-ray crystal analysis.

## Figures and Tables

**Figure 1 medicina-58-00463-f001:**
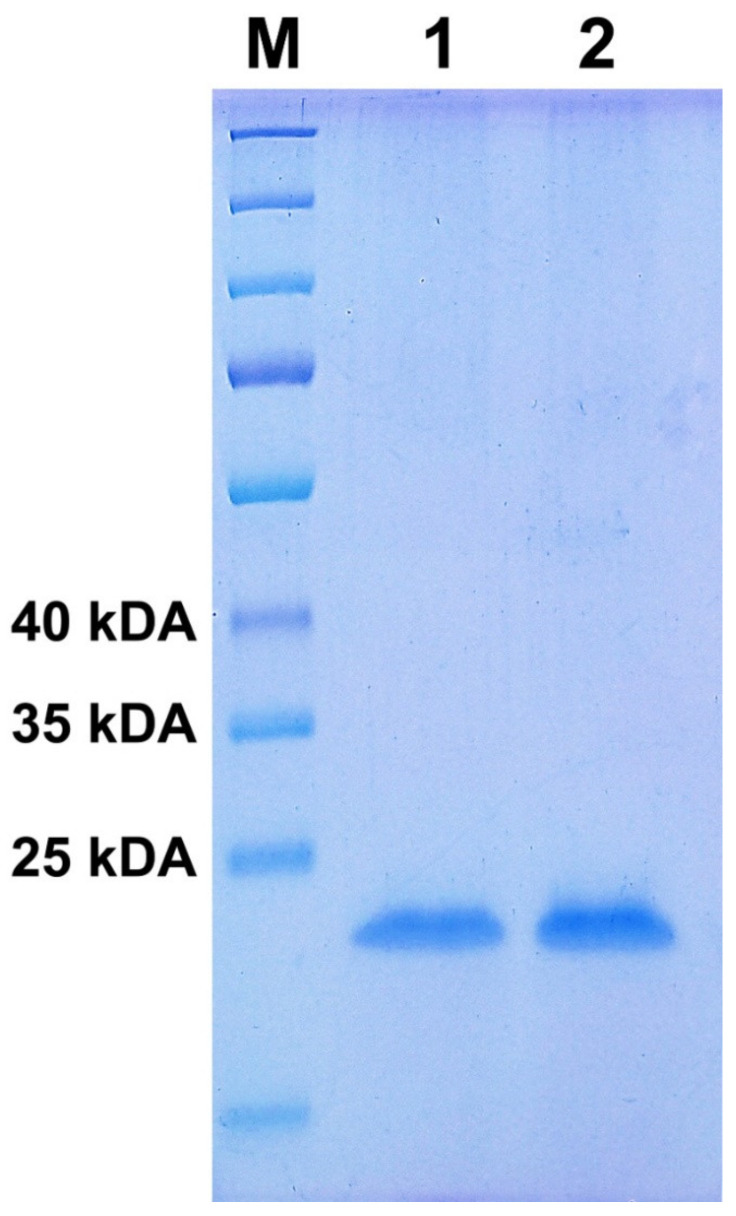
SDS page electrophoresis of IFNβ-1a 2.6 μg of Rebif (1–2) and ReciGen (3–4); M is a ladder from 25 kDa to 200 kDa.

**Figure 2 medicina-58-00463-f002:**
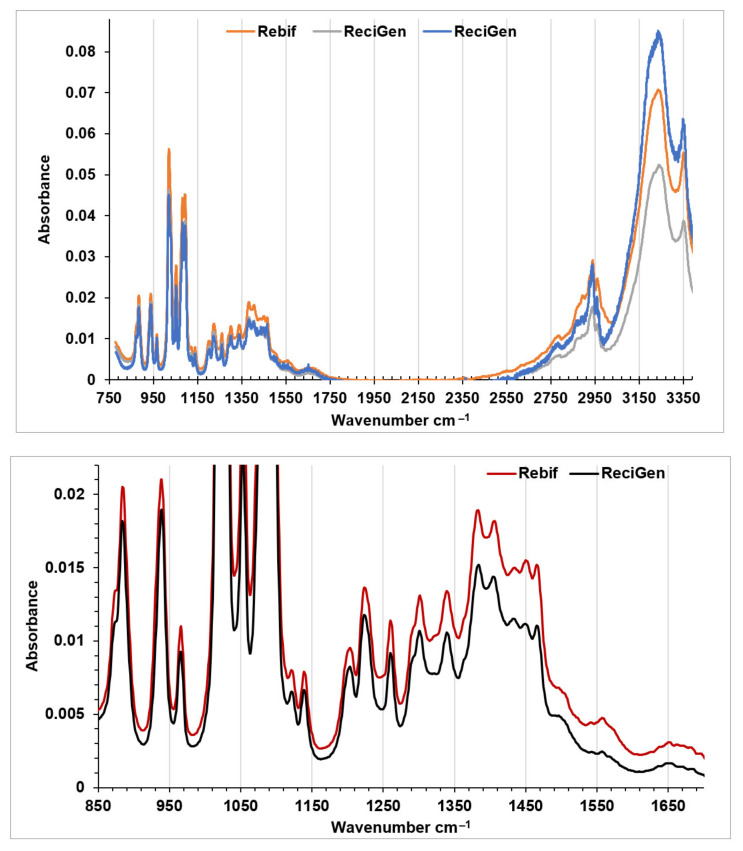
FTIR spectra of lyophilized solution of IFNβ-1 preparations in Rebif and ReciGen.

**Figure 3 medicina-58-00463-f003:**
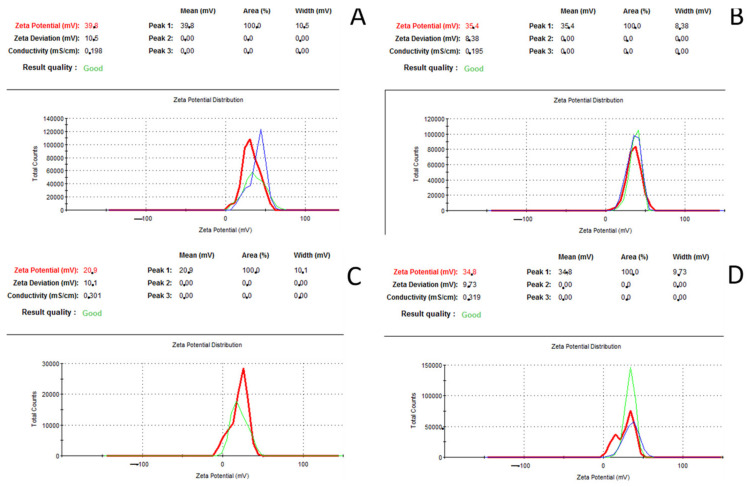
Zeta potential data of IFNβ-1a: (**A**,**B**) ReciGen; (**C**,**D**) Rebif.

**Figure 4 medicina-58-00463-f004:**
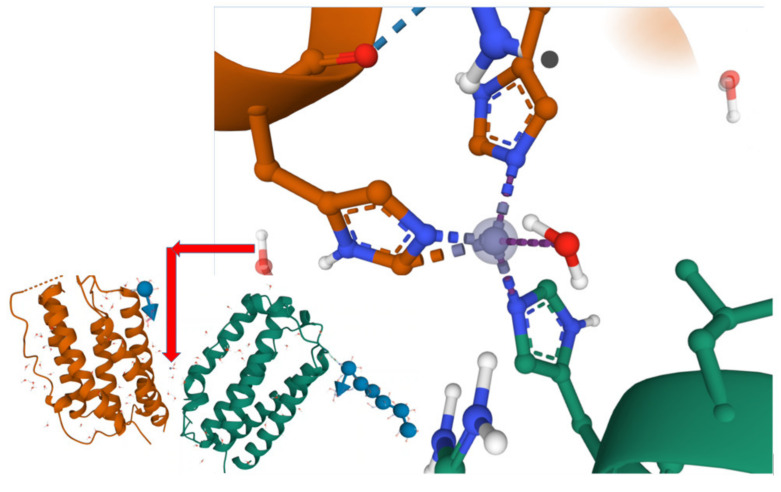
3D-structure of IFNβ-1 with highlighted Zn^2±^-complex the visualization was performed using software Mol* Viewer for 3D visualization and analysis of large biomolecular structures [37].

**Table 1 medicina-58-00463-t001:** Metal ion content in the of IFNβ-1a evaluated using ICP MS analysis.

Metal Ion	Content of IFNβ-1a						ICH Q3D (R1)PDE, μg/Day
	ReciGen, μg/L	ReciGen, in One Syringe, Weight of IFNβ-1a, ng	Accumulated Expose Dosage after ReciGen Application for One Year, μg	Rebif, μg/L	Rebif in One Syringe, Weight of IFNβ-1a, ng	Accumulated Expose Dosage after Rebif Application for One Year, μg	
Al	7.40 ± 0.17	3.70	0.58	ND	ND	ND	-
Cr	13.80± 0.32	6.90	1.08	7.11 ± 0.16	3.7	0.58	15.0
Mn	1.39 ± 0.03	0.70	0.11	ND	ND	ND	-
Fe	75.8 ±1.7	38.0	5.93	ND	ND	ND	-
Ni	4.63 ± 0.10	2.30	0.36	1.70 ± 0.04	0.85	0.13	20.0
Zn	2.88 ± 0.06	0.001	0.22	ND	ND	ND	-
As	2.69 ± 0.062	1.29	0.20	ND	ND	ND	15.0
Hg	1.05 ± 0.024	0.50	0.08	ND	ND	ND	3.0
Σ metal ion, nmol		0.8785			0.0857		

ICP MS, inductively coupled plasma mass spectrometry; ICH Q3D (R1), Harmonised guideline for elemental impurities Q3D (R1); PDE, Permitted Daily Exposure; ND, not detected.

**Table 2 medicina-58-00463-t002:** Specification of Rebif and ReciGen IFNβ-1a in a syringe with volume 0.5 mL.

**Pharmaceutical Active Ingredient**		
IFNβ-1a recombinant human	22.0 μg (6 million units) (19.8–24.2 μg)	44.0 μg (12 million units) (39.6–48.4 μg)
**Supplementary Substances (EurPh, USP)**		
Benzyl alhocol	2.5 mg	2.5 mg
Mannitol	22.5 mg	22.5 mg
Methionine	0.06 mg ± 0.01	0.06 mg ± 0.01
Poloxamer 188	0.25 mg	0.25 mg
0.01 M acetate buffer pH 4.2	0.5 mL	0.5 mL

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
