# Peer review of "Comparative Toxicity of Interferon Beta-1a Impurities of Heavy Metal Ions"

_medicina, 2022, doi:10.3390/medicina58040463_

Round 1
Reviewer 1 Report
It is a fascinating manuscript that broadens our scope by the impact of impurities on the toxicity of biological products. I made some suggestions that might increase the strength of the manuscript and/ or sharpen the manuscript.
The title should include Bioequivalences or toxicological equivalence, or comparative toxicity.
Remove from 36 - 45: Thus, … developments of severe side effects.
Remove from 53 – 56: Proteins …. Treatment of covid-19.
Remove 60 – 63: It is known that …. Described elsewhere
Also, the conflict-of-interest statement should include a clear message that the author doesn't intend to promote one brand over another and has no obligation and/or conflict with one of those companies.
Author Response
Dear reviewer,
I am greatly thankful to your valuable comments that certainly improve quality of the manuscript.
Comment "The title should include Bioequivalences or toxicological equivalence, or comparative toxicity."
I agree with the comment . The title of the manuscript was refined to"
Comparative toxicity of interferon beta-1a impurities of heavy metal ions
Comment Remove from 36 - 45: Thus, … developments of severe side effects.
The comment was addressed.
Comment Remove from 53 – 56: Proteins …. Treatment of covid-19.
The sentence was removed.
Comment Remove 60 – 63: It is known that …. Described elsewhere
The sentences were removed.
Comment Also, the conflict-of-interest statement should include a clear message that the author doesn't intend to promote one brand over another and has no obligation and/or conflict with one of those companies.
I totally agree with the comment . The statement was included to the text.
Reviewer 2 Report
The manuscript entitled “Heavy metal ions in impurity of interferon beta-1a contained drugs affect protein conformation” by Berillo Dmitriy is trying to use FTIR and ICP MS to analyze the residual heavy metal ions in the marketed IFN b-1a, Recigen and Rebief. The author concluded that 45.05% and 4.4% of the IFN b-1a protein in recigen and Rebief, respectively, is in a metal-bound or nonnative conformation. The work may give an evaluation of the drug quality. However, the experimental design, quality and the writing is not enough for publishing on the professional journals.
- The manuscript didn’t give clear and concise introduction, results, conclusions. I am difficult to understand what the author wants to express. A lot of unrelated descriptions appeared in the introduction, for example the market of protein therapeutics. The author needs to focus on why the metal ion contamination maters in protein therapeutics.
- When a value is shown, the standard error needs to be reported. The size of the samples also needs to mention in legend or in Method.
- The SDS gel in Figure 1 is not good enough to show impurities. High concentration of gel is needed. The loaded protein is too less, two to three time more is needed.
- The author claimed in row 215 that Figure 1 indicates the same isoelectric point of the samples. However, SDS gel is not able to report the isoelectric points.
- The author reported that Recigen IFN b-1a contains 0.88 nM heavy metal ions and claimed 45.05% are in different conformation. This is not possible. Normally, the metal ion-coordination residues His, Glu, Asp and Cys have affinity at um to mM range. Only highly specific metal ion binding pocket can bind with sub-nM affinity. Thus, your experimental results cannot support your conclusions.
- The 3 cm-1 difference in your FTIR spectra is not enough to claim a different conformation considering your wavelength resolution is 4 cm-1.
Author Response
I am very grateful to the reviewer for valuable comments.
Comment 1. The manuscript didn’t give clear and concise introduction, results, conclusions. I am difficult to understand what the author wants to express. A lot of unrelated descriptions appeared in the introduction, for example the market of protein therapeutics. The author needs to focus on why the metal ion contamination maters in protein therapeutics.
Reply. The introduction was modified and unrelated information removed.
Comment. When a value is shown, the standard error needs to be reported. The size of the samples also needs to mention in legend or in Method.
reply. I thank the reviewer for the valuable note. I apologies for the missing standard error. The comment was addressed in the text.
Comment. The SDS gel in Figure 1 is not good enough to show impurities. High concentration of gel is needed. The loaded protein is too less; two to three time more is needed.
Reply. We cannot increase the concentration as it is commercially available solution. I agree with the raised question and therefore larger volume of interferon samples were run via SDS gel. Additional analysis did not reveal noticeable impurities.
Comment. The author claimed in row 215 that Figure 1 indicates the same isoelectric point of the samples. However, SDS gel is not able to report the isoelectric points.
Reply. I apologies with the typo and the sentence was corrected accordingly. I have tried to estimate the size distribution of proteins using DLS using a Malvern Nano ZS (Malvern Instruments, Warriewood, UK), but the concentration of protein is a bit lower the limit of detection of the instrument and therefore the results were not included. Fortunately, I managed to obtain data about zeta potential for both interferons, that was incorporated to the manuscript.
Comment. The author reported that Recigen IFN b-1a contains 0.88 nM heavy metal ions and claimed 45.05% are in different conformation. This is not possible. Normally, the metal ion-coordination residues His, Glu, Asp and Cys have affinity at um to mM range. Only highly specific metal ion binding pocket can bind with sub-nM affinity. Thus, your experimental results cannot support your conclusions.
Reply. I thank the reviewer for the note. I removed 45% complex formation, I draw another conclusion and supported the statement by several research papers https://www.ncbi.nlm.nih.gov/pmc/articles/PMC2872550/ Zheng, H., Chruszcz, M., Lasota, P., Lebioda, L., & Minor, W. (2008). Data mining of metal ion environments present in protein structures. Journal of inorganic biochemistry, 102(9), 1765-1776.
Gamble, A. J., & Peacock, A. F. (2014). De novo design of peptide scaffolds as novel preorganized ligands for metal-ion coordination. Protein Design, 211-231.
Kozlowski, H., Potocki, S., Remelli, M., Rowinska-Zyrek, M., & Valensin, D. (2013). Specific metal ion binding sites in unstructured regions of proteins. Coordination Chemistry Reviews, 257(19-20), 2625-2638.
Comment. The 3 cm-1 difference in your FT-IR spectra is not enough to claim a different conformation considering your wavelength resolution is 4 cm-1.
Reply. I agree with the point and therefore additional experiments were carried out at higher resolution(0.5cm-1) . The figures were replaced and the discussion modified accordingly.
Reviewer 3 Report
The manuscript describes the comparison of metal contaminants in two pharmaceutical forms of IFNbeta-1a. The two products have significantly different metal contaminants, as well as some differences in protein secondary structure. These results suggest that there may be different activities of the two different preparations.
Major comments:
- Although there are large differences in the metal ions contained in the two IFNbeta-1a products, their presence does not necessarily mean that they are bound to the protein. The differences in the FTIR spectra are small, suggesting that there are some differences but the potential effects on function are unknown. Also, summing the total metal for calculation of percent of protein may overstate the effect, as not all metals detected will have the same effects on protein conformation and activity. Together, these lead to overstatement of the conclusions about the effect on IFNbeta-1a structure and function. These statements need to be revised or more experiments need be performed to strengthen them, such as CD, SAXS, or thermal stability analyses to look at protein conformation and functional assays.
- The legend for Fig 1 states that 44 ug of protein are loaded in each lane, whereas the methods state that 1.5 ug of protein are loaded. Which is correct? From the signal intensity, 1.5 ug would seem to be the amount loaded. However, this loading level may not be sufficient for detection of possible dimer or other protein bands. So, to assess dimers and protein contaminants, a larger amount of protein (20-30 ug) should be loaded.
- Why are the absorbances for the two FTIR spectra so different? Comparison of the two spectra would be easier if plotted in a single graph.
Author Response
Comment. Although there are large differences in the metal ions contained in the two IFNbeta-1a products, their presence does not necessarily mean that they are bound to the protein. The differences in the FTIR spectra are small, suggesting that there are some differences but the potential effects on function are unknown. Also, summing the total metal for calculation of percent of protein may overstate the effect, as not all metals detected will have the same effects on protein conformation and activity. Together, these lead to overstatement of the conclusions about the effect on IFNbeta-1a structure and function. These statements need to be revised or more experiments need be performed to strengthen them, such as CD, SAXS, or thermal stability analyses to look at protein conformation and functional assays.
Reply. Author thanks reviewer for valuable comments and appreciate time spent on the revision process. I agree with your point, Unfortunately I did not find CD and SAXS in our country. Certainly in following studies we will try to run the CD and SAXS. To support the statement of effect of metal ions on IFNbeta-1a structure and function zeta potential and dynamic light scattering measurement using Malvern Nano ZS (Malvern Instruments, Warriewood, UK) was performed. The concentration of protein in the sample is a bit lower the limit of detection of the instrument and therefore size distribution of proteins using DLS with good signal was not obtained. I obtained data about zeta potential for both interferons, which illustrated some difference. Additional discussion was added to the manuscript.
Regarding the complex formation between polyvalent metal ions and proteins there are a lot of references in the text that supports the hypothesis. Of cause in order to completely confirm the bioequivalence test on humans have to be performed, it requires significant funding and time.
McConnell, K. D., Fitzkee, N. C., & Emerson, J. P. (2022). Metal Ion Binding Induces Local Protein Unfolding and Destabilizes Human Carbonic Anhydrase II. Inorganic Chemistry.
https://www.ncbi.nlm.nih.gov/pmc/articles/PMC2872550/ Zheng, H., Chruszcz, M., Lasota, P., Lebioda, L., & Minor, W. (2008). Data mining of metal ion environments present in protein structures. Journal of inorganic biochemistry, 102(9), 1765-1776.
Gamble, A. J., & Peacock, A. F. (2014). De novo design of peptide scaffolds as novel preorganized ligands for metal-ion coordination. Protein Design, 211-231.
Kozlowski, H., Potocki, S., Remelli, M., Rowinska-Zyrek, M., & Valensin, D. (2013). Specific metal ion binding sites in unstructured regions of proteins. Coordination Chemistry Reviews, 257(19-20), 2625-2638.
- Sánchez-Alarcón, J., Milić, M., Bustamante-Montes, L. P., Isaac-Olivé, K., Valencia-Quintana, R., & Ramírez-Durán, N. (2021). Genotoxicity of Mercury and Its Derivatives Demonstrated In Vitro and In Vivo in Human Populations Studies. Systematic Review. Toxics, 9(12), 326. DOI: 10.3390/toxics9120326
- Lohren, H., et al., Toxicity of organic and inorganic mercury species in differentiated human neurons and human astrocytes. Journal of trace elements in medicine and biology, 2015. 32: p. 200-208. https://doi.org/10.1016/j.jtemb.2015.06.008
- Jacobson, T., et al., Cadmium causes misfolding and aggregation of cytosolic proteins in yeast. Molecular and cellular biology, 2017. 37(17): p. e00490-16. https://doi.org/10.1128/MCB.00490-16
- Chiarelli, R. and M.C. Roccheri, Heavy metals and metalloids as autophagy inducing agents: focus on cadmium and arsenic. Cells, 2012. 1(3): p. 597-616. https://doi.org/10.3390/cells1030597
- Radhakrishnan, R., et al., Zinc mediated dimer of human interferon-α2b revealed by X-ray crystallography. Structure, 1996. 4(12): p. 1453-1463. https://doi.org/10.1016/S0969-2126(96)00152-9
- Derrick, T.S., et al., Effect of metal cations on the conformation and inactivation of recombinant human factor VIII. Journal of pharmaceutical sciences, 2004. 93(10): p. 2549-2557. https://doi.org/10.1002/jps.20167
32 Crisponi, G., et al., Chelating agents for human diseases related to aluminium overload. Coordination Chemistry Reviews, 2012. 256(1-2): p. 89-104. https://doi.org/10.1016/j.ccr.2011.06.013
- Armstrong, R.A., S.J. Winsper, and J.A. Blair, Hypothesis: is Alzheimer's Disease a Metal-induced Immune Disorder? Neurodegeneration, 1995. 4(1): p. 107-111. https://doi.org/10.1006/neur.1995.0013
Comment. The legend for Fig 1 states that 44 ug of protein are loaded in each lane, whereas the methods state that 1.5 ug of protein are loaded. Which is correct? From the signal intensity, 1.5 ug would seem to be the amount loaded. However, this loading level may not be sufficient for detection of possible dimer or other protein bands. So, to assess dimers and protein contaminants, a larger amount of protein (20-30 ug) should be loaded.
Reply: Thanks for the valuable note, 1.5 ug was loaded for the SDS gel analysis. A new the SDS gel analysis was done at higher amount 2.63 ug of interferon. I will try to make the analysis at a larger amount of protein (20-30 ug) in nearest future. At the moment I run out of samples. I order several brands of INF-b on the market It can take some time.
Comment. Why are the absorbance’s for the two FTIR spectra so different? Comparison of the two spectra would be easier if plotted in a single graph.
Reply. Thanks for the comment, I performed the FTIR analysis using better precision equipment and plotted spectra in a single graph.
Round 2
Reviewer 3 Report
The author has addressed the concerns raised in the revised manuscript, with additional results included.
Author Response
We thank Reviewer for these overall comments. We have improved the manuscript with more details where suggested;
In a previous version I have obtained FTIR spectra at the resolution that was requested, additionally the text and conclusion in the manuscript was changed. I agree with the reviewer that conventional FTIR analysis is not sensitive at such low concentrations of proteins and moreover in the complex mixture.
"FTIR analysis confirms the presence of peptide in native form in both samples. More de-tailed information cannot be extracted from these spectra due to presence of supplementary substances such as acetate, methionine, poloxamer, mannitol because of overlay of signals of functional groups".
The conclusion was also adjusted
The IFNβ-1a Rebif contain approximately 4.5 mol. % of ions relatively the protein amount, but IFNβ-1a ReciGen contains approximately 10 fold more, which may be an acceptable level according to the ICH Q3D.
I appologies for some typo in the MS. Spelling of the manuscript was checked.
Best regards